# Trends of adult height in India from 1998 to 2015: Evidence from the National Family and Health Survey

**Krishna Kumar Choudhary**[ID]*, Sayan Das, Prachinkumar Ghodajkar

Centre of Social Medicine and Community Health, JNU, New Delhi, India

* krishnavagda@gmail.com

**Data Availability Statement:** The data used for this manuscript are publicly available and it can be accessed by formal request to the DHS Program after registration at the site. The authors chose India's data which is available according to the

## Abstract

### Aim

The aim of the study is to investigate the trends in adult height between two consecutive surveys of NHFS and explore differences across variables such as gender, wealth, social groups etc.

### Methods

We used the NFHS-II (1998–99), NFHS–III (2005–2006) and NFHS-IV (2015–16) (all three for women and last two for men) data to examine the trends in average height. Comparison was done between the two age strata of 15–25 and 26–50 years, across both male and female, to assess the trends.

### Results

Between NFHS-III and NFHS-IV, the average height of women in the age group of 15–25 showed a decline by 0.12 cm [95% CI, -0.24 to 0.00, p-0.051] while in the 26–50 years age strata it demonstrated significant improvement in the mean height by 0.13 cm [95% CI, 0.02 to 0.023, p-0.015]. However, Between NFHS III and IV, the average height of women in the poorest wealth index category registered a significant decline [-0.57cm, 95% CI, -076 to -0.37, p-0.000]. Between NFHS III and IV, the average height of Scheduled Tribe (ST) women in the age group of 15–25 years also exhibited a significant decline by 0.42 cm, [95% CI, -0.73 to -0.12, p-0.007]. Among men, between the two surveys, both the age groups of 15–25 years and 26–50 years showed significant decline in average height: 1.10 cm [95% CI, -1.31 to -.099 cm, p-0.00] and 0.86 cm [95% CI, -1.03 to -0.69, p-0.000], respectively.

### Conclusion

In the context of an overall increase in average heights worldwide the decline in average height of adults in India is alarming and demands an urgent enquiry. The argument for different standards of height for Indian population as different genetic group needs further scrutiny. However, the trends from India clearly underline the need to examine the non-genetic

survey rounds. NHFS-4 (DHS-VII) has a file named "IAPR74FL.DTA" for men and "IAIR74FL" for women. NFHS-3 (DHS-V) has the file "IAIR52FL" for women and "IAMR52FL" for men. NFHS-2 (DHS-IV) has only women data and the file name is "IAIR42FL". The data can be accessed through the DHS site (The DHS Program - login_main) after successful registration and approval from the competent authority.

**Funding:** The authors received no specific funding for this work.

**Competing interests:** The authors have declared that no competing interests exist.

factors also to understand the interplay of genetic, nutritional and other social and environmental determinants on height.

## Introduction

Height is widely accepted and recognised as one of the key measures of physical well-being and biological standard of living at the population level [1–3]. In addition to genetic potential, other factors such as socio-economic condition, disease history, access to quality health services, and nutritional security have a significant bearing on human stature [2,4]. The role of nutrition over stature, however, has had a long and contested history among nutritionists, policymakers and health professionals. In India, the debate was recently fueled by Dr. Panagariya's argument [5] on stunting, and subsequent critiques of it by various scholars [6–11]. Evidently, much of this scholarship on stunting and height has focused on children. Little attention has been paid to adult height, a more or less constant phenomenon having a higher potential for exploration of determinants.

Genetic, environmental and social factors, all influence the height attained by an individual. Although it is claimed that genetic factors determine 60–80% of the final height, environmental and social factors contribute significantly to the realisation of that potential [12,13]. The genetic potential here refers to an optimum level of height achievable in ideal conditions. Individual's realization of the given genetic potential is a reflection of food intake, as well as availability, accessibility and effectiveness of health care service during the growing period [14]. It is well established that the attained height of an individual is an embodiment of nutritional status and other determinants, traced from womb to late adolescence [14].

Several studies in the field of economics, health and nutrition demonstrate the relationship between Gross Domestic Product (GDP) [15], per capita expenditure on food, net nutritional intake disease occurrence etc. and stature [16–21]. Adequate nutrition is closely linked with an individual's achievement of his/her genetic potential of height growth. The impact of nutrition on height gain can start as early as fetal life [22–24]. Malnutrition during infancy, especially stunting around puberty significantly affects the final height gain in adulthood [15,25]. On the other hand, overall change in factors such as nutrition, sanitation, hygiene, transport and income have been found to positively impact height and weight gain among Indian school children [26]. The international study of childhood poverty found mid-day meal program to have improved both stunting and underweight [27]. Deficits in final height or the failure of realization of genetic potential of a population ultimately reflect prior nutritional stress which as the examples show is conditioned by social and environmental factors over time [13].

Socio-economic factors like household characteristics (number of siblings, occupation, class and locality etc.) have a bearing on human stature [28]. Caste is another social factor that shapes different socio-economic determinants and consequent inequality across different aspects of life in India [29]. Data from National Family and Health Survey 4 show strong and significant correlation between average adult height and social category. The average height of people belonging to Scheduled Tribe (ST) and Scheduled Caste (SC) category were found to be shorter than Other Backward Class (OBC) and others. Moreover, across all these categories, the richest in the wealth index had the highest mean height while the poorest had the lowest, clearly signaling the role of class [30]. Analysis of NFHS III data show higher socioeconomic status to be associated with greater height and greater secular increase in height [31]. Insofar as

gender is concerned, over time the average height of Indians for both male and female have improved but the improvement was higher in male compared to female [32].

Despite considerable evidence to the contrary, the role of genetic factors has often been used to downplay the role of food intake, standard of living and health care, especially in countries like India. For instance, Panagariya, explaining stunting among Indian children, argued that the shorter stature of Indian children is due to their low genetic potential, and the higher proportion of malnutrition, otherwise considered a causal factor, is a result of anthropometric error. The role of socioeconomic and environmental factors were left out of the equation. Also was overlooked the dynamic relation between genetic potential and socio-economic and environmental conditions, evolving over a long period of time, ultimately expressed as height of an individual or average height of a population group [5]. There are very few primary data based studies done in the Indian context that explain the relationship of height attainment with the explanatory variables [33–35], Most of the work on height has been done in the developed countries. Thus, the phenomenon of average height of an adult population demands further exploration in the Indian context.

To explore these issues, the paper has attempted to analyze the height trends and patterns in relation to a number of selected variables such as sex, religion, types of caste or tribe, place of residence, wealth index, geography etc., using data from the National Family and Health Survey (NFHS-II,III,IV) rounds.

## Methodology

To trace the trends of height among adults in India, quantitative secondary data analysis was employed to explore the variation in height across the explanatory variables. The data on height on a large scale is collected under the erstwhile National Nutritional Monitoring Bureau (NNMB)and the National Family Health Survey (NFHS). The NNMB collected the height data of Indian children and adults from 1974. The survey was, however, limited to only 11 states (Andhra Pradesh, Gujarat, Karnataka, Kerala, Madhya Pradesh, Maharashtra, Orissa, Tamil Nadu, Uttar Pradesh and West Bengal) of India. Hence the NNMB rounds may be representative sample of states, but it cannot completely represent the country. The NFHS, on the other hand, is a representative sample survey with a larger sample size than NNMB. Therefore, we have chosen NFHS-II (1998–99), III (2005–06), IV (2015–16) rounds data to see the trends and pattern of adult height among Indians. The height of adult was measured as per the Demographic and Health Surveys (DHS) manual [36,37] by using portable stadiometer in NFHS-IV. Rest of the survey manual were not reported by the surveying agency. However, in the report of NFHS-III it was mentioned that it is measured as health investigator manual and in NFHS-II investigator were trained in the anthropometry measurement at AIIMS and IIPS. NFHS for all its rounds of surveys had obtained written consent from the participants. The authors had taken written permission from DHS to use that data for relevant analysis in the present paper. The data used was de-identified and anonymized by NFHS already.

The data on adult women's height was collected by NFHS from the second round onwards, while for men, it was in collected only in the last two rounds. Therefore, the NFHS II, III, and IV rounds were used for women's height analysis and NFHS III and IV rounds for analysing men's height. From all the NFHS rounds, age between 15 to 50 years (for women 15 to 49) was considered for the analysis. Two strata of age groups were created to assess the trends, 15 to 25 years, and 26 to 50 years. The rationale behind this stratification is that in the available literature, growth in height has been documented until the age of 20–25 [38–40]. NFHS sample is stratified into two stage sample and census 2011 is used as sampling framework for the selection of primary sampling unit (PSU) for the village and census enumeration block (CEB) for

the urban ward [41]. The samples drawn for analysis of women's height were 83876 out of 90303 from NFHS-II, 121728 out of 138592 from NFHS-III, and 700602 out of 749344 from NFHS-IV. For men's height, sample of 66468 out of 74396 from NFHS- III and 105783 out of 126543 from NFHS-IV were drawn. Bivariate analysis was carried out. A linear regression model was run with the adjustment of the clusters to calculate the t-test value for identifying the relationship between variables from the two consecutive rounds of the survey, and to find whether the difference is significant or not.

## Result

### Women

**15–25 years age group.** The mean height of Indian women in the age group of 15–25 years showed significant improvement between NFHS-II and NFHS-III as indicated by the calculated coefficient of 0.84 cm [95% CI, 0.69 to 0.99, p-0.000]. However, between NFHS-III and NFHS-IV, the average height of women in this age group showed a decline by 0.12 cm [95% CI, -0.24 to 0.00, p-0.051], albeit not significant (S1 Table, Fig 1).

From NFHS-II to NFHS-III, the average height of women showed significant improvement among all castes and all age groups. However, the improvement in the average height of tribal women was not significant (S4 Table). Between NFHS-III and IV, Tribal women in the age group of 15–25 years, showed a significant decline in their average height by 0.42 cm, [95% CI, -0.73 to -0.12, p-0.007] (S5 Table).

From NFHS II to NFHS III, women across all places of residence showed significant improvement in their average height (S7 Table). From NFHS-III to NFHS-IV, the average height of women in the poorest wealth index category was observed to have suffered a significant decline [-0.63 cm, 95% CI, -0.87 to -0.44, p-0.000]. On the other hand, women who belonged to richer wealth index showed significant improvement in their mean height [0.22 cm, 95% CI, 0.02 to 0.42, p-0.029] (S8 Table).

The average height of women from NFHS-II to NFHS-III, across different states, demonstrated significant improvement in this age group. However, Meghalaya showed a significant decline in women's height growth of about 2.27 cm [95% CI, -3.80 t -0.73, p-0.004]. Except for Kerala, Mizoram, Odisha, Punjab, and Sikkim, the improvement was significant in the rest of the states.

From NFHS-III to NFHS-IV, Kerala, Mizoram, and Sikkim showed significant improvement in women's height in the age group of 15 to 25 years while Tamil Nadu [-0.65 cm, 95% CI, -1.19 to -0.12, p-0.017] and Haryana [-0.82 cm, 95% CI, -130 to -0.35, p-0.001] showed significant decline. Sikkim showed the highest improvement in women's height of about 1.25 cm [95% CI, 0.49 to 2.02, p-0.001]. (Fig 2, S10 Table).

**26–50 years age group.** Between NFHS II and III, women in this age group, like their younger counterparts, registered a significant improvement (0.55 cm) in mean height [95% CI, 0.43 to 0.67, p-0.000]. Between NFHS III and IV, however, this group showed significant improvement by 0.13 cm [95% CI, 0.02 to 0.023, p-0.015], unlike the 15–25 age group.

Except for Christians [-0.19, 95% CI, -0.77 to 0.40, p-0.527], women across other religious groups experienced improvement in their mean height between NFHS II and NFHS III. From NFHS III to IV, Christian and Buddhist /neo-Buddhist women registered significant improvement in their average height by 0.58 cm [95% CI, 0.10 to 1.06, p- 0.017] and 1.11 cm [95% CI, 0.37 to 1.85, p-0.003], respectively. (S2 Table).

Between NFHS-III and IV, the average height of women across all castes, in the age groups of 26–50 years, improved significantly except that of the women from scheduled tribes. Women belonging to scheduled caste [0.50cm, 95% CI, 0.31 to 0.69, p-0.000] and other

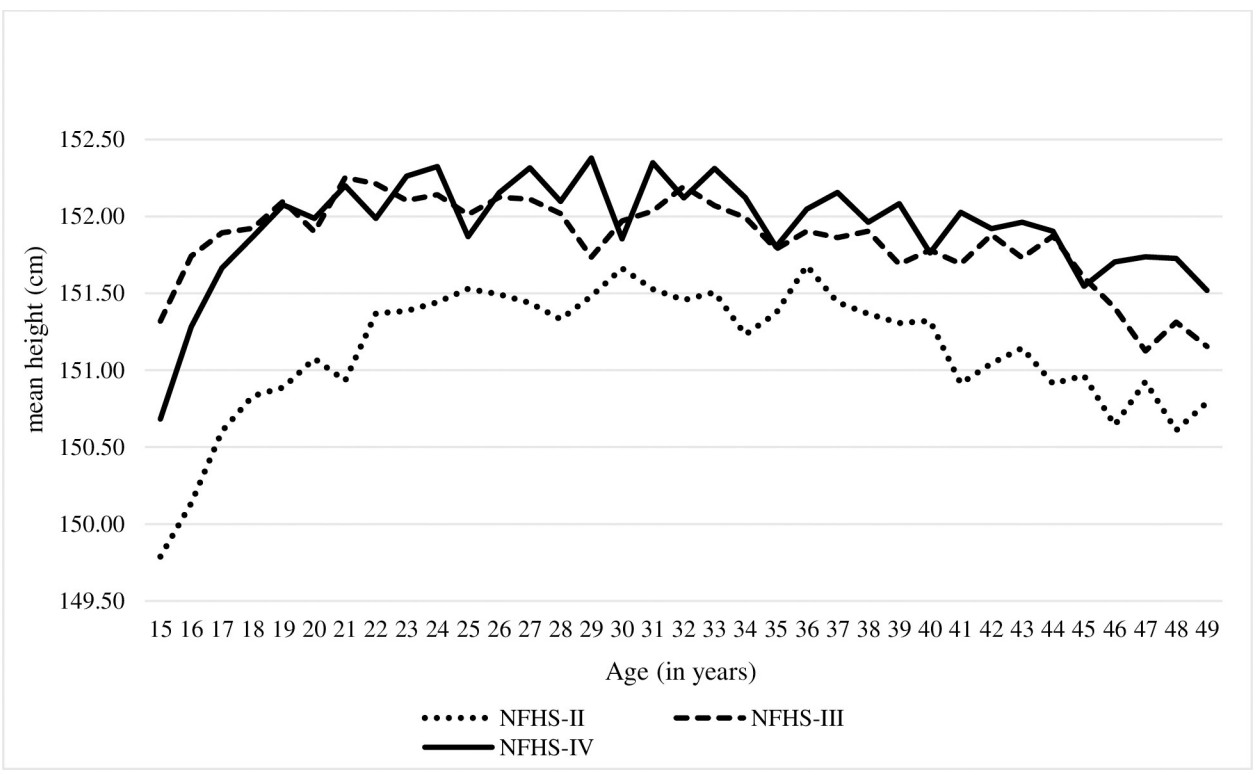

**Fig 1. Distribution of mean height of women according to the age.** From NFHS-II to NFHS-III round, women across religious groups experienced improvement in their mean height. This is significant among Hindu, Muslim and Sikh women (S3 Table). From NFHS-III to IV, women of 15–25 years, belonging to Sikh community, exhibited a significant decline in mean height by 0.66 cm [95% CI, -1.16 to -0.16, p-0.009] while among Jain community, women showed significantly improved mean height by 1.72 [95%CI, 0.05 to 3.38, p- 0.044]. Buddhist/neo-Buddhist women were also observed to have improved their average height by 0.40 cm [95% CI, -0.70 to 1.49, p- 0.478]. (S2 Table).

backward class category [0.23 cm, 95% CI, 0.08 to 0.38, p- 0.002] showed significant improvement in their mean height.

Women across all places of residence in this age group showed significant improvement in average height between NFHS II to III. From NFHS-III to NFHS-IV, the improvement in height was significantly better among women in urban area [0.20 cm, 95% CI, 0.03 to 0.38, p-0.024] than their rural counterparts [0.06 cm, 95% CI, -0.07 to 0.18, p- 0.379] (S6 Table).

Between NFHS III and IV, the average height of women in the poorest wealth index category registered a significant decline [-0.57cm, 95% CI, -076 to -0.37, p-0.000], just as their younger counterpart. Women from the middle [0.20cm, 95% CI, 0.03 to 0.37, p-0.020], richer [0.51cm, 95% CI 0.34 to 0.67, p-0.000] and richest [0.31 cm, 95% CI, 0.14 to 0.49, p-0.000] wealth index, however, showed significant improvement in the age group of 26–50 years.

In the age group of 26 to 50 years, except Meghalaya, all states showed improvement in the average height of women between NFHS II and III. (Fig 3, S9 Table). Jammu and Kashmir [1.11 cm, 95% CI, 0.66 to 1.55, p-0.000], Kerala [2.51cm, 95%CI, 2.09 to 2.93, p-0.000], Mizoram [0.57 cm, 95% CI, 0.04 to 1.09, p-0.034], Punjab [1.14cm, 95% CI, 0.74 to 1.53, p-0.000] and Sikkim [1.88 cm, 95% CI, 1.28 to 2.48, p-0.000] showed significant improvement in mean height of women between NFHS III and IV. Bihar [-0.48 cm, 95%CI, -0.83 to -0.14, p-0.006], Madhya Pradesh [-0.64cm, 95% CI, -0.92 to -0.37, p-0.000], and Delhi [-2.46cm, 95% CI, -3.35 to -1.58, p-0.000], on the other hand, showed significant negative trends in the average height of women (Fig 2, S10 Table).

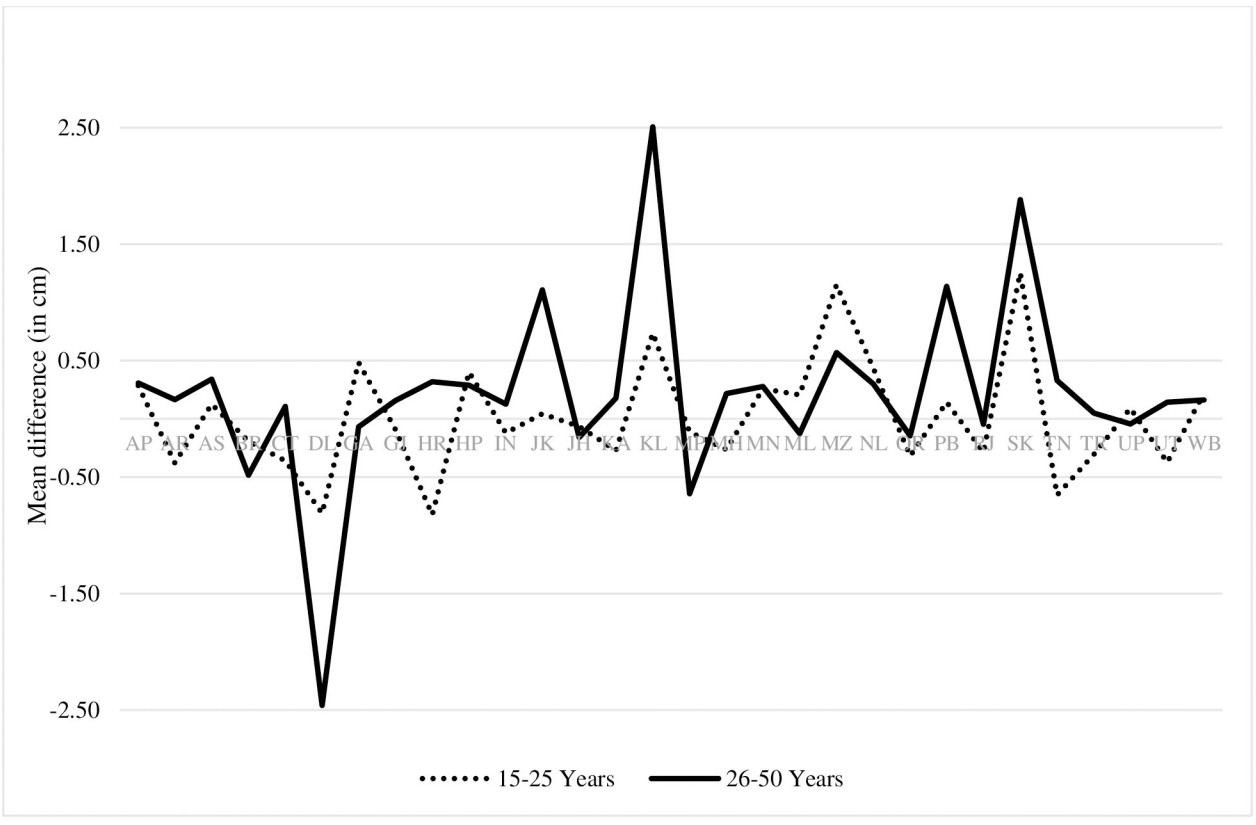

**Fig 2. Women's height trends between NFHS-III and NFHS-IV.**

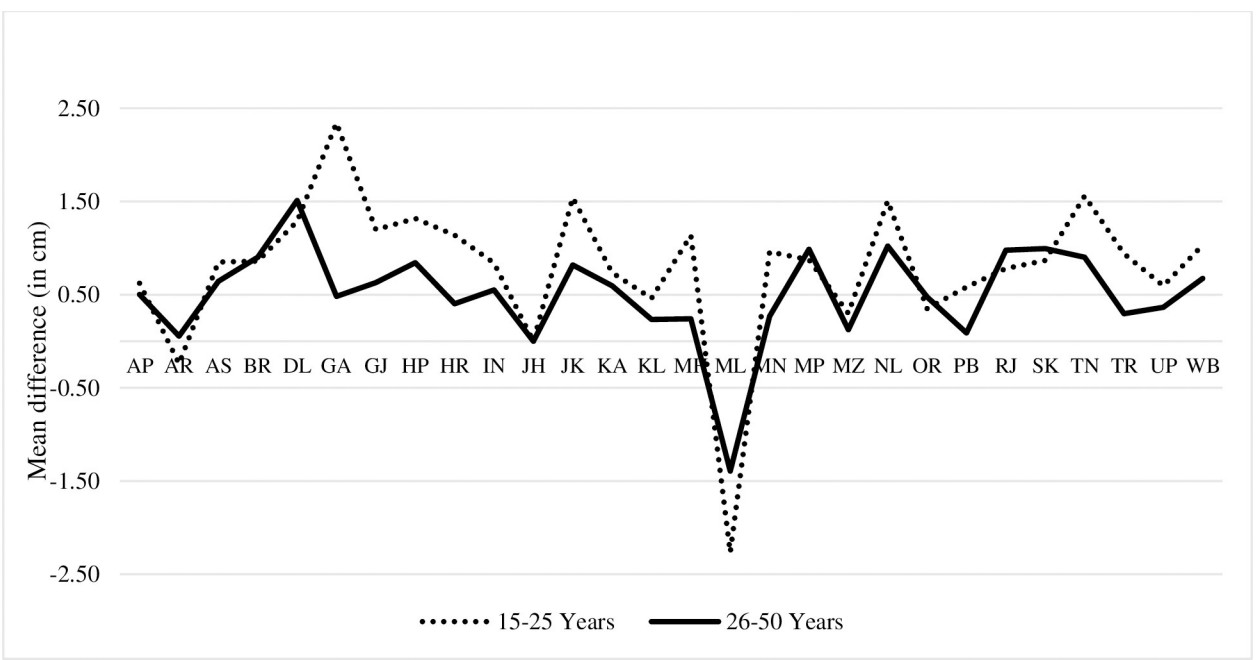

**Fig 3. Women's height trends between NFHS-II and NFHS-III.**

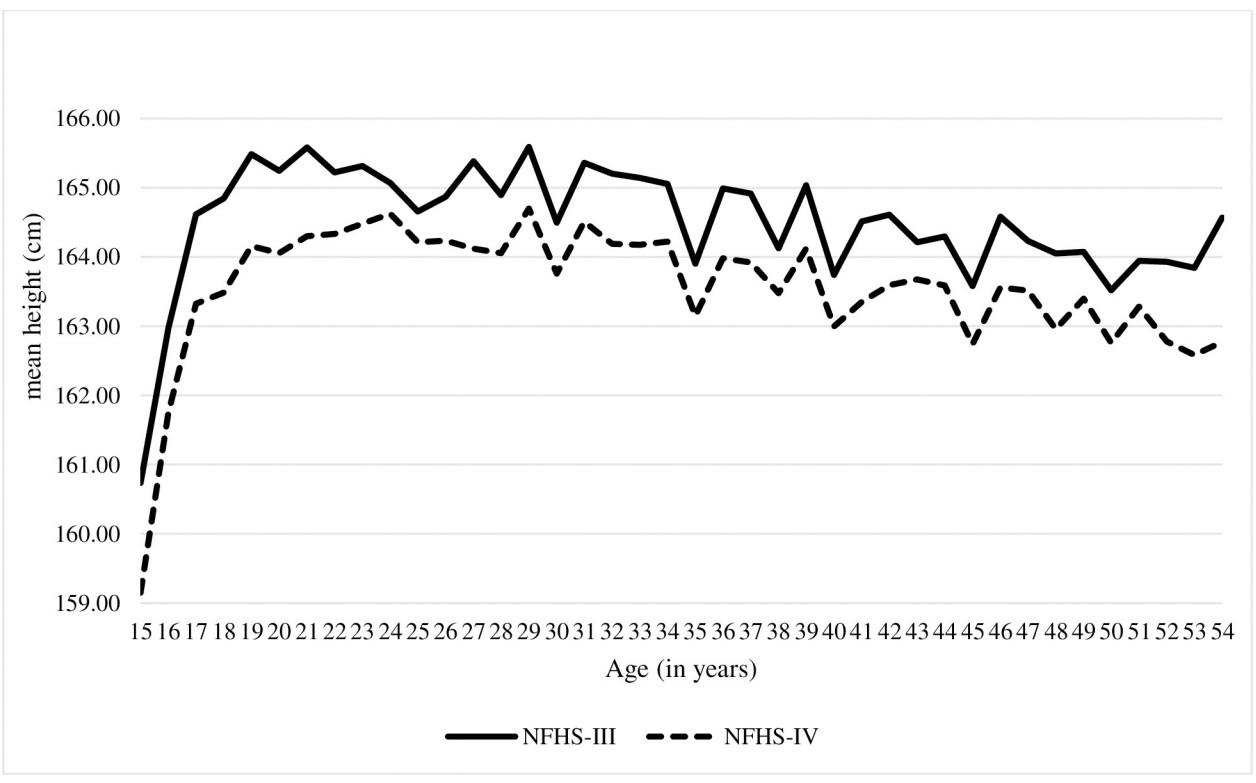

**Fig 4. Distribution of mean height of men according to the age.**

## Men

**15–25 years age group.** The average height of Indian men has significantly declined by 1.10 cm [95% CI, -1.31 to -.099 cm, p-0.00], between the NFHS-III and IV (S1 Table, Fig 4). The decline in average height was observed across religious group, caste or tribe, and residence, wealth Index (S2, S5, S6 and S8 Tables).

The state-wise trends of average height of men in the age group of 15 to 25 years find only men from Nagaland showing significant improvement in their average height [1.27 cm, 95% CI, 0.20 to 2.34, p-0.02]. Himachal Pradesh, Manipur, Meghalaya, and Mizoram also showed improvement in the average height of men while rest of the states exhibited decline in the average height of men (S11 Table, Fig 5).

**26–50 years age group.** Between NFHS-III and IV, the average height of Indian men has significantly declined by 0.86 cm [95% CI, -1.03 to -0.69, p-0.000] (S1 Table, Fig 4). Similarly, decline was observed in average height of men across caste or tribe, religious group, wealth index and place of residence (S1, S5, S6 and S8 Tables). The decline was more pronounced in the urban area.

In the age group of 26–50 years, average height of men from Himachal Pradesh registered significant and greatest improvement by 1.06 cm [95% CI, 0.21 to 1.91, p-0.015]. The highest decline, about 2.04 cm, was observed in Karnataka [95% CI, -2.76 to 1.33, p-0.00]. Arunachal Pradesh, Assam, Bihar, Gujarat, Haryana, Jharkhand, Karnataka, Madhya Pradesh, Maharashtra, Delhi, Odisha, Punjab, Rajasthan, Uttar Pradesh, and Uttarakhand, all were found to have had significant decline in the average height of men (Fig 5).

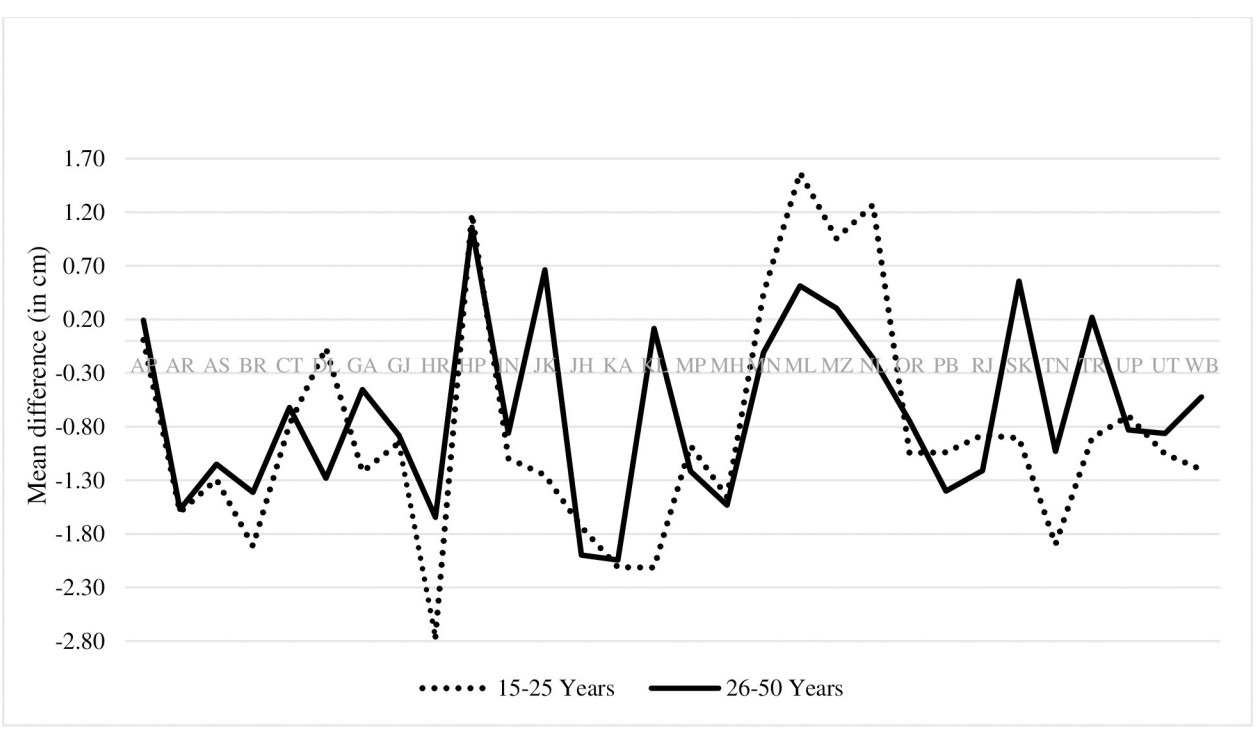

**Fig 5. Men's height trends between NFHS-III and NFHS-IV.**

## Discussion

The analysis reveals some worrying trends concerning the average adult height in India. While between NFHS II and III, the average height of women across age groups improved significantly, it showed a decline between NFHS III and IV in the age group of 15 to 25 years. If disaggregated across different social groups, tribal women seem to have persistently suffered the most negative outcome. Their average height, between NFHS II and III, did not register any significant increase, while it improved significantly for the rest of the women. Between NFHS III and IV, the average height of all women in the age group of 26 to 50 years improved significantly, except for ST women who exhibited a decline. And when average height of women in the age group of 15–25 years declined between the last two rounds of NFHS, decline in height was the most significant and also the highest among women from the scheduled tribe. Women from the poorest wealth index category also suffered a similar fate, when between NFHS III and IV their height registered a significant decline across both the age groups of 15–25 years and 26–50 years. Interestingly, women in the age group of 15 to 25 years in NFHS IV, whose height declined compared to the same age group of women from NFHS III, come from the post-90s birth cohorts, the period when neoliberal policies gained momentum in India. For men, the findings revealed less of variance, with average height declining significantly between NFHS III and IV across all age groups, castes, tribes and residence. Curiously, maximum significant decline was observed among both the poorest and the richest wealth index categories.

The decline in height brings the role of various socioeconomic determinants of height back in focus. The Scheduled Tribes, constituting 9% of India's population as per 2011 census, represent one of the most economically and educationally disadvantaged groups in India. Majority of them live in rural and isolated areas with poor government services. While decline in disparities in wages, employment and consumption between SC and STs and the general caste

is debatable, there's enough evidence to suggest large sections from the SC and STs continue to face discrimination and deprivation. STs often face exclusion from government services and suffer displacement from productive land [42]. Researchers analyzed NFHS III data to show that an average five-year-old ST girls is 2 cm shorter than an average general caste girl. They further found the differences in socioeconomic status to be responsible for the entire gap in height between ST and general caste children [43]. Multiple regression analysis from the NFHS data can also perhaps unearth such explanatory linkages between various socioeconomic determinants and decline in height of adult ST women across both age groups in NFHS IV.

There is, however, considerable research on childhood undernutrition. Many of our nutritional programs are also designed to address that. But the role of adolescent nutrition on adult height gain is less explored. Adolescence is the second time in human life with rapid physical growth, during which children gain 50% of adult weight and 20% of adult height [44]. Under the Integrated Child Development Services, the Ministry of Women and Child Development implements the adolescent girls' scheme which provides nutritional support to 11 to 14-year-old girls. But there's no similar program for adolescent boys. Research on dietary intake and nutritional status during different periods of life for both Indian men and women can therefore be useful in searching for answers.

Undoubtedly, this decline in the average adult height of different groups among men and women is a matter of utmost concern and demands inquiry into non-genetic determinants of height, irrespective of the relative role of genetic factors in deciding height. Admittedly, the results may not present a completely coherent picture, but they compel us to further examine the role of social and economic factors in determining adult height. And while economic and nutritional status of Indian population is showing overall improvement, these trends in height also raise questions about equitable distribution of these benefits across the population.

## Conclusion

We believe, in the context of an overall increase in average heights worldwide the decline in average height of adults in India is alarming and demands an urgent enquiry. The argument for different standards of height for Indian population as different genetic group also needs further scrutiny. The important questions would be- What is the genetic potential for height in Indian population? Is it different for other countries or population groups? Are we achieving that genetic potential? What factors shape realization of the given genetic potential? And here, we feel the trends from India clearly underline the need for examining the non-genetic factors to understand and find solutions to these disturbing trends.

## Limitations

While research has shown that adult height is a function of environmental and social factors that determine the realization of genetic potential of different population groups, there are very few studies in the Indian context that delineate the pathways through which that happens. Available literature has established the causal relation of height with different social-environmental factors but not the processes. That is also a limitation of the current study. This study only analyses available country level data to observe trends in adult height over the different rounds of NFHS, hoping future research picks up on the findings to study their linkages with different determinants and the processes shaping them.

## Supporting information

**S1 Table. Distribution of mean height of Indian according to the age group.**
(DOCX)

**S2 Table. Distribution of mean height of men and women according the age group and religion, rounds NFHS-3 and NFHS-4.**
(DOCX)

**S3 Table. Distribution of mean height of women according to religion, rounds NFHS-2 and NFHS-3.**
(DOCX)

**S4 Table. Distribution of mean height of women according to the type of caste or tribe, rounds NFHS-2 and NFHS-3.**
(DOCX)

**S5 Table. Distribution of mean height of men and women according to the type of caste or tribe, round NFHS-3 and NFHS-4.**
(DOCX)

**S6 Table. Distribution of mean height of men and women according to the residence, NFHS-4 and NFHS-3.**
(DOCX)

**S7 Table. Distribution of mean height of women according to the residence, NFHS-2 and NFHS-3.**
(DOCX)

**S8 Table. Distribution of mean height of men and women according to the wealth index, NFHS-4 and NFHS-3.**
(DOCX)

**S9 Table. State wise distribution of mean height of women according to age group, NFHS-3 and NFHS-2.**
(DOCX)

**S10 Table. State wise distribution of mean height of women according to age group, NFHS-4 and NFHS-3.**
(DOCX)

**S11 Table. State wise distribution of mean height of men according to age group, NFHS-4 and NFHS-3.**
(DOCX)

## Acknowledgments

We thank to Dr. Srinivas Goli and Ashish Gupta for their guidance in the data analysis.

## Author Contributions

**Conceptualization:** Krishna Kumar Choudhary, Sayan Das, Prachinkumar Ghodajkar.

**Data curation:** Krishna Kumar Choudhary.

**Formal analysis:** Sayan Das.

**Methodology:** Krishna Kumar Choudhary, Prachinkumar Ghodajkar.

**Supervision:** Prachinkumar Ghodajkar.

**Writing – original draft:** Krishna Kumar Choudhary, Sayan Das.

**Writing – review & editing:** Prachinkumar Ghodajkar.

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
