## [Decision Letter · Decision Letter 0]

5 May 2021

PONE-D-21-05757

Trends of adult height in India from 1998 to 2015: Evidence from the National Family and Health Survey

PLOS ONE

Dear Dr. Choudhary,

Thank you for submitting your manuscript to PLOS ONE. After careful consideration, we feel that it has merit but does not fully meet PLOS ONE’s publication criteria as it currently stands. Therefore, we invite you to submit a revised version of the manuscript that addresses the points raised during the review process.

Reviewers are in favour of recommending this piece. However, reviewers suggest some minor revisions to the manuscript. I recommend authors to consider those minor suggestions. 

We look forward to receiving your revised manuscript.

Kind regards,

Srinivas Goli, Ph.D.

Academic Editor

PLOS ONE

Additional Editor Comments:

Reviewers are in favour of recommending this piece. However, reviewers suggest some minor revisions to the manuscript. I recommend authors to consider those minor suggestions.

Journal Requirements:

3. Please include your tables as part of your main manuscript and remove the individual files. Please note that supplementary tables should be uploaded as separate "supporting information" files.

5. We note that you have reported significance probabilities of 0 in places. Since p=0 is not strictly possible, please correct this to a more appropriate limit, eg 'p<0.0001'.

Reviewers' comments:

Reviewer's Responses to Questions

**Comments to the Author**

1. Is the manuscript technically sound, and do the data support the conclusions?

Reviewer #1: Yes

Reviewer #2: Yes

2. Has the statistical analysis been performed appropriately and rigorously? 

Reviewer #1: Yes

Reviewer #2: I Don't Know

3. Have the authors made all data underlying the findings in their manuscript fully available?

Reviewer #1: Yes

Reviewer #2: No

4. Is the manuscript presented in an intelligible fashion and written in standard English?

Reviewer #1: Yes

Reviewer #2: No

5. Review Comments to the Author

Reviewer #1: I would like to appreciate authors for this intensive research work and robust analysis of nationally representative data on the most discussed topic such as “Trends of adult height in India from 1998 to 2015: Evidence from the National Family and Health Survey.” The presentation of the data seems empirically strong and the paper is generally well written, with an appropriate focus. This study may be accepted for publication in the journal with reviewer following suggestions and comments for further improvements.

1. The authors can incorporate the Limitations of the study in the separate section of the manuscript.

Reviewer #2: The manuscript entitled ‘Trends of adult height in India from 1998 to 2015: Evidence from the National Family and Health Survey’ has well written using recent rounds of nationwide demographic and health data. The manuscript has sufficiently elaborated past research and clearly mentioned the objectives and have done robust analysis. However, I have some comments to authors to consider while revising the manuscript.

1. Introduction: Much of introduction focused on how genetic factors contribute to the height of population. However, other social-demographic and nutrition role in determining the height is not well explained. Further, authors must address to provide the research gap.

2. Methods: Authors must provide the sampling strategy employed by the NFHS. Also give more detailed information about how height of adult population has been measured. Further, provide the delayed clarification if any different methods used to measure the height in different rounds of NFHS survey.

3. Discussion: Discussion section is need to be strengthened with current literature on height of population with regards to various biological and social studies. Also, write a separate section on conclusion.

6. PLOS authors have the option to publish the peer review history of their article (what does this mean?). If published, this will include your full peer review and any attached files.

Reviewer #1: No

Reviewer #2: **Yes: **Suresh Jungari

---

## [Author Response · Author response to Decision Letter 0]

15 Jun 2021

Reviewer-1

Suggestions- The authors can incorporate the Limitations of the study in the separate section of the manuscript.

Response- A separate section on ‘Limitations’ have been added after ‘Conclusion’

Reviewer-2 

Suggestions-

1. Introduction: Much of introduction focused on how genetic factors contribute to the height of population. However, other social-demographic and nutrition role in determining the height is not well explained. Further, authors must address to provide the research gap.

Response- The role of socio-demographic factors and nutrition in determining height has been incorporated in the introduction section (3rd and 4th paragraph in the introduction section) Research gap has also been addressed in the introduction section (in the penultimate paragraph of the introduction section)

2. Methods: Authors must provide the sampling strategy employed by the NFHS. Also give more detailed information about how height of adult population has been measured. Further, provide the delayed clarification if any different methods used to measure the height in different rounds of NFHS survey.

Response- The information sought has been incorporated within the two paragraphs of the Methodology section.

3. Discussion: Discussion section is need to be strengthened with current literature on height of population with regards to various biological and social studies. Also, write a separate section on conclusion.

Response- Two new paragraphs have been incorporated in the discussion sections to address the points as suggested. A separate section on Conclusion has been included.

---

## [Editor Report · Decision Letter 1]

21 Jun 2021

PONE-D-21-05757R1

Trends of adult height in India from 1998 to 2015: Evidence from the National Family and Health Survey

PLOS ONE

Dear Dr. Choudhary,

Thank you for submitting your manuscript to PLOS ONE. After careful consideration, we feel that it has merit but does not fully meet PLOS ONE’s publication criteria as it currently stands. Therefore, we invite you to submit a revised version of the manuscript that addresses the points raised during the review process.

ACADEMIC EDITOR: Before sending back to re-review, I request authors to write the abstract according PLOS One authors guidelines. PLOS accepts a structured abstract. Please see headings required in structured abstract from PLOS One author guidelines. Also, check formatting of other headings, Tables and its headings. 

We look forward to receiving your revised manuscript.

Kind regards,

Srinivas Goli, Ph.D.

Academic Editor

PLOS ONE

Journal Requirements:

Additional Editor Comments (if provided):

Before sending back to re-review, I request authors to write the abstract according PLOS One authors guidelines. PLOS accepts a structured abstract. Please see headings required in structured abstract from PLOS One author guidelines. Also, check formatting of other headings, Tables and its headings.

---

## [Author Response · Author response to Decision Letter 1]

20 Jul 2021

Abstract, heading, sub heading and table heading were edited as suggested. Furthermore, all the annexure tables are included as supporting information (tables).

---

## [Editor Report · Decision Letter 2]

22 Jul 2021

Trends of adult height in India from 1998 to 2015: Evidence from the National Family and Health Survey

PONE-D-21-05757R2

Dear Dr. Choudhary,

We’re pleased to inform you that your manuscript has been judged scientifically suitable for publication and will be formally accepted for publication once it meets all outstanding technical requirements.

Kind regards,

Srinivas Goli, Ph.D.

Academic Editor

PLOS ONE

Additional Editor Comments (optional):

Authors have done all suggested revisions and the manuscript is ready for publication in PLOS One.
---

## [Editor Report · Acceptance letter]

1 Sep 2021

PONE-D-21-05757R2 

Trends of adult height in India from 1998 to 2015: Evidence from the National Family and Health Survey 

Dear Dr. Choudhary:

I'm pleased to inform you that your manuscript has been deemed suitable for publication in PLOS ONE. Congratulations! Your manuscript is now with our production department. 

Kind regards, 

on behalf of

Dr. Srinivas Goli 

Academic Editor

PLOS ONE